# Robotic-Assisted Simple Prostatectomy: A Systematic Review

**DOI:** 10.3390/jcm9061798

**Published:** 2020-06-09

**Authors:** Yakup Kordan, Abdullah Erdem Canda, Ersin Köseoğlu, Derya Balbay, M. Pilar Laguna, Jean de la Rosette

**Affiliations:** 1Department of Urology, School of Medicine, Koc University, Zeytinburnu, 34010 Istanbul, Turkey; ecanda@kuh.ku.edu.tr (A.E.C.); ekoseoglu@kuh.ku.edu.tr (E.K.); dbalbay@ku.edu.tr (D.B.); 2Department of Urology, School of Medicine, Istanbul Medipol University, 34083 Istanbul, Turkey; m.p.lagunapes@gmail.com (M.P.L.); j.j.delarosette@gmail.com (J.d.l.R.); 3Amsterdam University Medical Centers, 1105 Amsterdam, The Netherlands

**Keywords:** benign prostate hyperplasia, miminally invasive simple prostatectomy, minimally invasive adenomectomy, robotic adenomectomy, robotic simple prostatectomy

## Abstract

Contemporary minimally invasive surgical (MIS) treatment options of patients with male Lower Urinary Tract Symptoms (LUTS) in men with prostate glands >80 mL include Holmium Laser Enucleation Prostate (HoLEP), Thulium laser VapoEnucleation Prostate (ThuVEP), and Laparoscopic (LSP) or Robotic-Assisted Simple Prostatectomy (RASP). Implementing new laser technologies is costly, and the steep learning curve of these laser techniques limit their wide range use. This promoted the use of LSP and RASP in centers with readily established laparoscopy or robotic surgery programs. The aim of this study is to review case and comparative series of RASP. We systematically reviewed published data from 2008 to 2020 on RASP and have identified 26 non-comparative and 9 comparative case series. RASP has longer operation time but less time spent in hospital and less blood loss. The outcomes of improvements in symptom score, post-voiding residual urine (PVR), postoperative PSA decline, complications, and cost are similar when compared to open and laser enucleation techniques. These outcomes position RASP as a viable MIS treatment option for patients with male LUTS needing surgical treatment for enlarged prostates. Nevertheless, prospective, randomized controlled trials (RCTs) with multicenter and large sample size are needed to confirm the findings of this systematic review.

## 1. Introduction

Male lower urinary tract symptoms (LUTS) caused by an enlarged prostate adversely affects many men and impairs their quality of life. Despite adequate medical therapy, surgical interventions are recommended in men with urinary retention, impaired renal function, and dilatation of upper urinary tract secondary to obstruction, recurrent urinary tract infections, recurrent hematuria, accompanying bladder stones, diverticulas, and refractory symptoms. 

The “gold standard” surgical intervention for large prostate glands > 80 gr has traditionally been open simple prostatectomy (OSP). Retropubic and suprapubic (transvesical) OSP are the most common and widespread techniques for large adenomas, especially in developing countries. However, OSP is usually associated with substantial perioperative complications and morbidity of up to 42%, including prolonged catheterization time, increased estimated blood loss (EBL), length of hospital stay (LOS), and a transfusion rate of more than 24% [1]. Therefore, a variety of minimally invasive surgical techniques have emerged and extensively investigated for the treatment of large obstructing adenomas. 

The first laparoscopic (LSP) was reported in 2002 by Mariano et al. [2]. Since then, LSP was quickly adopted and widely used by urologists experienced in laparoscopy. In conjunction, published series of LSP increased every year until 2008 when the first report of robotic-assisted simple prostatectomy (RASP) was published in 2008 by Sotelo et al. [3]. 

The recent data on laser technologies such as holmium laser enucleation prostate (HoLEP) and thulium laser vapoenucleation prostate (ThuVEP) established these techniques as alternative approaches to OSP [4]. This has been adopted in two major guidelines recommendations. For large prostate glands >80 mL, American Urological Association (AUA) guidelines [5] recommend HoLEP, ThuVEP, or simple prostatectomy (SP) which can be performed in open, laparoscopic, or robotically according to the surgeon’s expertise and discretion. On the other hand, the European Association of Urology (EAU) guidelines [6] recommend HoLEP, bipolar enucleation, and open simple prostatectomy (OSP) as first line treatment options. However, LSP and RASP, which are minimal invasive simple prostatectomy (MISP) techniques, are still regarded as feasible in men with prostate sizes >80 mL needing surgical treatment in EAU guidelines. EAU guidelines also indicate the need of randomized controlled studies for further recommendation. An increasing body of evidence supports the role of HoLEP and ThuVEP in the treatment of BPH with large prostate >80 gr given they are “size independent” treatment approaches with favorable long-term outcomes [4,7]. However, some of these treatment options also enharbour several limitations. For example, to become sufficient to perform HoLEP, the medical procedure requires new equipment to perform such a surgery and has a steep learning curve. Several studies concluded that approximately 50 cases were needed for an efficient HoLEP surgery [8,9]. These limitations have led to the search for alternative new energies for enucleation and alternative surgical techniques. With the increasing use of robotic surgery in urology, several teams have explored the option to perform a robotic assisted simple prostatectomy (RASP) for large prostate glands. Robotic systems are widely distributed, and RASP is easy to implement and perform. RASP presents all the advantages of minimal invasiveness of LSP. Moreover, RASP provides three-dimensional vision, five degrees of movement, faster learning curve, especially regarding suture techniques, and comfortable operating position overcoming disadvantages of LSP. An increasing body of evidence confirms that RASP offers comparable outcomes to open and laser enucleation techniques for improvement in outcomes at a favorable low complication rate and costs [4]. Besides, RASP, compared with OSP, provides decreased length of hospital stay, blood loss, and transfusion prevalence [10].

Detailed scrutiny of the present literature revealed that there is no randomized controlled trial (RCT) which compares RASP with OSP or other treatment modalities. Recent systematic reviews and metanalyses compared MISP (RASP + LSP) with OSP or other endoscopic techniques such as lasers, vaporizations, and bipolar resections. However, there is not any recent systematic review or a metanalysis directly comparing RASP to other treatment modalities. This systematic review of RASP critically analyzes the current data available and provides recommendations for the use of RASP in clinical practice and possible implementation in guidelines.

## 2. Methods

### 2.1. Evidence Acquisition

We performed a structured, comprehensive review of the current literature for RASP. Eligible articles were identified through the electronic databases PUBMED, MEDLINE, and Web of Science. PRISMA guidelines were followed to select relevant papers. Figure 1 shows the article selection process according to PRISMA guidelines. The search employed the term “Benign prostatic hyperplasia” AND “minimally invasive simple prostatectomy” OR “robotic simple prostatectomy” OR “robotic adenomectomy”. In addition, references in the reference sections of the identified publications were also added to the list. The raw data for this systematic review is publicly available through the Open Science Framework [11].

### 2.2. Inclusion and Exclusion Criteria

Original articles reporting outcomes of case series and comparative studies of RASP, other minimally invasive techniques (HoLEP, Green Light laser enucleation, ThuVEP etc.), or OSP were eligible. Review articles, case reports, and case series with less than three patients were excluded. Search was not limited by region or language. However, articles published in English between 2008 (the first published RASP article) and February 2020 were included in this analysis. Each article’s title, abstract and (when necessary) full articles were reviewed for their appropriateness and their relevance with regards to RASP by two authors (Y.K. and A.E.C.) and a third author (D.B.) resolved eventual discrepancies. Twenty-six original non-comparative and nine comparative case series were finally included in the evidence synthesis. 

### 2.3. Data Extraction and Outcome Measures

Data variables of interest were age, surgical approach and technique, preoperative and postoperative maximum flow rates (Qmax), preoperative and postoperative International Prostate Symptom Score (IPSS), operative time (OT), estimated blood loss (EBL), postoperative day of catheter removal, length of hospital stay (LOS), complications, incontinence and erectile dysfunction (International Index of erectile function (IIEF) and Sexual Health Inventory for men (SHIM)). Complications like blood transfusion, urinary tract infection, urinary retention, and conversion to OSP were carefully analyzed. Clavien–Dindo classification of complications were also included.

## 3. Results

We identified 573 relevant articles at our initial search. After omitting duplicates, articles not pertaining to RASP, and articles in languages other than English, 78 articles remained. 43 articles which include case reports, case series with <3 patients, reviews and meta-analyses, letters to editor, video presentations, and descriptions of techniques were removed from the study. Thirty-five studies were found to be eligible (26 case and 9 case comparative studies) for final analysis (Figure 1).

All 26 case studies [3,10,12,13,14,15,16,17,18,19,20,21,22,23,24,25,26,27,28,29,30,31,32,33,34,35] included in the final analysis were non comparative case series with level 3 evidence. In nine comparative studies [36,37,38,39,40,41,42,43,44], outcomes of RASP were compared to outcomes of other minimally invasive modalities (OSP, HoLEP, ThuVEP) for the treatment of benign prostate hyperplasia (BPH). Unfortunately, there have been no randomized clinical trials among these studies conducted to date. When the outcomes of RASP patients from these comparative studies are added to case studies the total number of RASP patients operated were 1564 (min: 3 patients and maximum: 487 patients) from 2008 to 2020 (Table 1). The surgical approach was transperitoneal in 24 studies and extraperitoneal in 6 series (Table 1). All RASP cases were operated by da Vinci 4 arm Surgical Systems (Intuitive, Sunnyvale, CA, USA), except for Fareed et al. [18] and Steinberg et al. [34] who reported their initial experience with 8 and 10 cases of RASP using the da Vinci Single Port surgical systems (Intuitive, Sunnyvale, CA, USA). While Fareed et al. [18] performed their surgeries through a GelPort (Applied Medical, Rancho Santa Margarita, CA, USA) inserted directly into the bladder, Steinberg et al. [34] inserted their single port system extraperitoneally and performed their surgeries.

In all these studies, the mean operation time varied from 90 minutes to 274 minutes. The weight of the adenoma specimen varied between 46.4 gr and 301 gr (Table 1). The mean foley catheter removal time ranged from 1.6 days to 13 days. The mean hospital length of stay varied from 1 day to 8.8 days. In most series, although there were no intraoperative complications, perioperative complications up to 37.5% [18] were reported (Table 1). Estimated blood loss varied between 98 mL and 558 mL. Transfusion rates were low. While in most series there was no need for blood transfusion, Fareed et al. [18] reported 87.5% transfusion rate in their study. The need of intraoperative conversion to open surgery was reported only in three studies [15,37,39]. Among them, only one study reported the reason of conversion to OSP. This was due to excessive bleeding (500 mL) and failure to progress during a RASP operation for a 260 gr prostatic adenoma [15]. 

Bladder neck contracture (BNC) was reported only in nine of these studies [12,13,22,28,37,40,42]. Although Yuh et al. [12] reported in one patient out of three and John et al. [13] in one patient out of 13, other researchers reported BNC in less than 1%. Sorokin et al. [42] compared RASP and OSP and reported BNC in two and zero patients among OSP and RASP groups, respectively. Autorino et al. [37] in their comparative study reported BNC in three out of 487 patients (0.6%) and three out of 843 patients (0.35%) in RASP and LSP groups, respectively.

Urinary incontinence (UI) was reported in eight studies [15,21,22,23,34,38,40,44]. Although Nestler et al. [44] reported UI in nine out of 35 RASP patients (25.7%), Elsamra et al. [23] in three out of 15 patients (20%), and Steinberg et al. [34] in one out of 10 patients (10%), in other series UI was less than 5.5% and mostly around 1%. However, these UI rates are early UI rates with less than a 3-month follow-up. 

Only 10 studies [15,21,22,28,30,31,32,37,38,39] reported on erectile function. In most of these studies, sexual function remained the same or improved (Table 1).

There are four studies comparing RASP outcomes to OSP [36,42,43,44,45] (Table 2). In all of these studies, the length of operation was significantly longer in RASP compared to OSP. On the other hand, RASP was found to have lower intraoperative EBL and shorter LOS. In addition, RASP was found to achieve similar functional outcomes, transfusion, and complication rates.

Similarly, in three studies [37,38,39] RASP was compared to LSP. While two studies [38,39] reported surgical and functional outcomes with no significant difference for EBL, catheter removal time, LOS, major complications rate between two groups, Autorino et al. [37] reported shorter operative time for LSP but less EBL in favor of RASP. 

There were two studies [40,41] comparing HoLEP to RASP and one study [44] comparing ThuVEP to RASP. While Umari et al. [40] reported similar improvements for Qmax, post-voiding residual urine (PVR), IPSS, similar operative time, and complication rates (no BNC observed in both groups), Zhang et al. [41] reported shorter mean operative time, catheter removal time, LOS, and lower transfusion rates in favor of HoLEP. Complication rates were also similar in this study. 

Nestler et al. [44] reported a matched pair analysis results of comparison for OSP, ThuVEP, and RASP. In their study, ThuVEP showed significantly lower operation time, blood loss, and transfusion rates over OSP (*p* < 0.01). However, although operation time was significantly lower in ThuVEP than RASP, the difference in favor of ThuVEP for blood loss and transfusion rates were not statistically significant (*p* = 0.18 and *p* = 0.36, respectively). Postoperative Qmax, IPSS, and QoL questionnaires improved without significant difference between the approaches (*p* > 0.08). In their study, only five patients needed one pad during the first 24 h after ThuVEP and median pad use was zero in ThuVEP. After RASP, 26 out of 35 patients reported no incontinence. Of the remaining nine patients, six patients needed one pad, two patients needed two pads, and one patient needed three pads per day. The difference between these two groups however was not statistically significant. In OSP, median pad use was 1 and mean pad use was 1.23, but overall 28 out of 35 patients needed pads, 14 patients one pad, 12 patients two pads, and 2 patients three pads. Mean as well as median pad use were significantly higher in OSP compared to the minimal invasive approaches (*p* ≤ 0.001). 

## 4. Discussion

Since 2008, when RASP was first described by Sotelo et al. [3], the procedure has been increasingly performed worldwide. Parallel to the increase in the centers implementing this technique, the literature on RASP also showed a steady growth with encouraging results [46]. RASP can be classified as transcapsular or transvesical. The approach can be transperitoneal or extraperitoneal. The cystostomy incision can be made horizontal or vertical. Recently Castillo et al. [28] reported a modified technique with longitudinal capsular and vesical incision, plication of prostatic capsule, and a posterior only urethrovesical anastomosis. Clavijo et al. [20] developed and reported a new “intrafascial technique” for RASP which was developed on the same principles of intrafascial radical prostatectomy technique. They performed near-complete excision of all prostatic tissue with preserving the seminal vesicles, periprostatic fascia, and puboprostatic ligaments. A subsequent vesicourethral anastomosis was performed in a manner similar to that of robotic prostatectomy. They claimed that this technique minimizes postoperative hematuria and has the advantage of not requiring postoperative bladder irrigation. Martin Garzon et al. [38] analyzed 236 minimally invasive simple prostatectomies. They reported 82 laparoscopic and 79 RASP with Millin approach and 75 RASP with intrafascial approach. They found similar surgical outcomes including complication rate, transfusion rate, and functional outcomes for IPSS, SHIM, continence, and Qmax between three procedures. They also concluded that an intrafascial technique does not require postoperative irrigation. Cacciamani et al. [29] reported RASP with 360° circumferential reconstruction and Wang et al. [30] published “Urethra Sparing” RASP series based on “Madigan surgical technique” first described in 1990 by Dixon et al. [47]. Recently, Simone et al. [32] reported, “Urethra and ejaculation preserving RASP” by implementing near-infrared fluorescence imaging-guidance to Madigan technique. They reported a 1-yr follow-up, median IPSS score, IIEF score, and MSHQ-EjD Short Form score as 5 (IQR 4–8), 26 (IQR 26–28), and 12 (IQR 1–14), respectively. They achieved satisfactory anterograde ejaculation in 66% of their patients. 

Although there is no prospective randomized study in the current literature, the data of existing case series (Table 1) and comparative studies (Table 2) suggest that RASP is a safe and efficacious procedure in the treatment of BPH patients with >80 gr of prostate size. In these studies, RASP reported to have longer operation time than OSP, LSP, and laser enucleations. However, RASP achieved similar functional outcomes for IPSS, PVR, Qmax, QoL, EBL, LOS, transfusion, and complication rates. Parsons et al. [45] studied the Nationwide Inpatient Sample (NIS) from 1998 to 2010 and identified 34,418 and 193 patients undergoing OSP and MISP, respectively. They focused on 2008 to 2010 because of the trend shift observed (increase in MISP and decrease in OSP) in surgical techniques during the last years. In their final analysis, 6027 OSP and 182 MISP patients were evaluated. They reported that MISP patients were more likely to have higher Charlson comorbidity scores and less likely to undergo transfusion, but these differences did not attain significance. They also found no significant differences in median LOS, hospital charges, or unadjusted in-hospital mortality. Similarly, Lucca et al. [48] performed a systematic review and meta-analysis comparing outcomes of MISP with OSP. They included 764 patients and 27 observational studies in which 8 were robotic surgery with 119 patients. They reported similar improvements in Qmax, IPSS in both groups while MISP group had significantly lower LOS, length of catheter use and EBL than OSP. However, the duration of operation was significantly lower in OSP. Authors, unfortunately, did not analyze RASP, LSP, and OSP subgroups separately. Li et al. [49] also conducted a systematic review and meta-analysis with 10 studies including 995 patients. Six of the studies were comparisons between LSP and OSP, while four were comparisons between RASP and OSP. They found no significant differences between MISP and OSP regarding postoperative IPSS, QoL, Qmax, PVR, and irrigation time. MISP group was found to have lesser EBL, shorter catheterization time, shorter LOS, lower transfusion rate, and lower complication rates compared with OSP group. They also reported longer operation time in MSP group. Moreover, they performed a subgroup analysis between RASP, LSP, and OSP to detect the possible differences. They demonstrated that OT, EBL, LOS, transfusion rate, and the number of complications in the LSP group was less than that in the OSP group (OR: 0.52; 95% CI: 0.37 to 0.74; *p* = 0.002), and the RASP group had the same result (OR: 0.42; 95% CI: 0.23 to 0.77; *p* = 0.005). These results were in conjunction with the results of our systematic review. 

Autorino et al. [37] collected data from 23 centers worldwide and reported the outcomes of largest RASP series to date. They reported 487 cases with median operative time of 154 min, median EBL of 200 mL, and median 2 days LOS. The time to foley removal was 7 (5–9) days in the study. However, since most patients were discharged from the hospital with their foley catheters and were removed on outpatient follow-up visits, the catheter removal time was not a good indicator for the evaluation of RASP outcomes. Similarly, blood transfusion rates were also not a good indicator for evaluation of RASP outcomes since they vary among institutions, according to patients’ comorbidities and surgeons’ preference. 

Umari et al. [40] compared 81 RASP to 45 HoLEP procedures and reported no difference in terms of operative time and complications. On the other hand, Zhang et al. [41] compared outcomes of 32 RASP and 600 HoLEP and reported shorter operative time in favor of HoLEP (103 min (std:47) versus 274 min (std: 49) (*p* < 0.001)). They collected HoLEP data from different centers. Johnson et al. [33] conducted a study to determine the learning curve for RASP and reported that 10-12 cases may be required for somebody with robotic experience to become proficient for RASP. On the other hand, Brunckhorst et al. [50] estimated 40–60 cases to be enough for HoLEP surgery. The authors concluded that learning curve might have an impact and favor RASP over HoLEP. Nestler et al. [44] reported significantly shorter operation time in ThuVEP than RASP (83 min versus 182 min, respectively). Regarding discrepancies in operating times among RASP series, Johnson et al. [33] also reported that operative time improved at a rate of 1 to 2 minutes per case, which represented the gaining experience of the surgery. Besides, the previous total robotic surgeries and robotic lower urinary system operations, like radical prostatectomies, cystectomies, and robotic systems (4 arm si or xi Da Vinci or single port system Da Vinci ), might have an impact on RASP operation time as well. Leonardo et al. [51] recently published a systematic review including 9 randomized clinical trials with most of them reporting data at 1 year to assess the management of patients with big prostates (>80cc). The investigated trials compared enucleation, vaporization, open techniques such as transurethral resection in saline (TURis), transurethral vaporization in saline (TUVis), bipolar plasma enucleation of prostate (BPEP), plasma kinetic resection of prostate (PKRP), transurethral vaporization of prostate (TVP), transurethral enucleation and resection of prostate (TUERP), plasma kinetic enucleation of prostate (PKEP), photo vaporization of prostate (PVP), diode enucleation of prostate (DILEP), HoLEP, ThuVEP, and OSP. In terms of perioperative outcomes, all the techniques had similar operative times and resected prostate weight, whereas, catheterization time and hospital stay were better in endoscopic techniques compared to open surgery. In terms of functional outcomes (IPSS, QMAX, and PVR), none of the techniques were proven superior to the other. When considering complications, open procedures carried a higher risk of transfusions, while no technique was proven superior to the others in terms of transient urge UI, BNC, and reintervention. 

BNC was reported as 3.0% ± 0.2% (between 0% and 6%) in OSP series [52]. This systematic review showed that BNC was reported <1% in most published RASP series. Although current data does not allow to metaanalyze the results of these studies because of uniformity of surgical techniques and data itself, one may speculate that BNC occurs less when compared to OSP and LSP. There is no data for comparing RASP with laser techniques as well. Leonardo et al. [51], in their recent systematic review, found the rate of BNC was between 15% and 75% in OSP and other endoscopic enucleation techniques, and they stated that none of the studies was designed to prove differences in this term. Similarly, UI was reported <5.5% in most RASP series. However, follow-up was very short and type of incontinence (stress versus urgency) was not specified in most series. The rate of urge UI was reported between 5% and 30% in OSP, endoscopic, and vaporization techniques [51]. Unfortunately, none of the current studies specifically addressed the comparative data of RASP and laser surgeries on the incidence of irritative symptoms that might be seen especially after laser surgeries in early postoperative period. Regarding erectile functions, current data does not allow to make direct comparison between RASP, LSP, OSP, and other endoscopic techniques. At least in most of the studies, which report on erectile or sexual function, postoperative results seemed unchanged or even increased in some extent. Long term RCT’s are needed to shed light on these gray zones. 

Transurethral resection of prostate (TUR-P) was widely used for the treatment of symptomatic BPH even for >80 gr prostates. In the current literature, there were studies comparing effectiveness between OSP and bipolar TUR-P [53]; however, to date, there was no study which compares the outcomes of bipolar TUR-P and RASP for the treatment of prostate glands >80 gr. One might speculate that bipolar TUR-P is the real competitor of the RASP even in cases of high prostatic volume, primarily due to the quicker and similar amount of tissue resection. Reviewing the literature, Zhu et al. [54] in 52 out of 132 patients who underwent bipolar TUR-P with preoperative transrectal ultrasound measured prostate volume of >80 mL (mean 101.6 mL) and reported that the resected BPH tissue in that specific subgroup was 64.75% of the initial measured prostate volume. Similarly, Matei et al. [19] reported in 25 patients with the mean (range) transrectal ultrasound (TRUS) prostate volume of 93.4 (70–150) mL, and the mean volume of tissue removed was 63.8 mL, accounting for 68.3% of the initial prostate volume. When they stratified patients into three subgroups, patients with prostate volumes of <80 mL, 81–99 mL, and >100 mL, the percentage of prostatic tissue removed decreased from 71.2% to 69% to 66.5%, respectively. Ou et al. [55] compared TUR-P and OSP and reported similar results in their series of 69 patients where only 53.2% of prostatic tissue was resected in the TURP group compared with 84.4% in the OSP group. Bach et al. [56] compared the efficacy of TUR-P, ThuVEP, and Greenlight PVP in 2648 patients with BPH. They reported efficacy, as measured in resection weight per minute of total operating time, was higher in ThuVEP than in TUR-P, independent of prostate volume. The tissue removal per operating time was higher in ThuVEP even in prostates <40 cc (15.4%), and this efficiency increased with increasing prostate volume to 51.6% in prostates larger than 80 cc. On the other hand, Ou et al. [53] in a prospective randomized study with 98 BPH patients compared the effectiveness of TUERP to OSP for prostate volumes of >80 mL. They reported the resected adenoma weight in the OSP group was more than that in the TUERP group, but the difference was not statistically significant (*p* = 0.062). When postoperative PSA reduction was used as a surrogate marker of adequacy of tissue removal, at 12 months postoperatively, a decrease in PSA was 72.9% (from 5.9 to 1.6 ng/mL) and 78.6% (from 5.6 to 1.2 ng/mL) in patients who underwent TUERP and OSP group, respectively. The mean postoperative PSA reductions in each group were similar (*p* = 0.12). The mean weight of the adenoma specimen removed in RASP series varied between 46.4 gr and 301 gr and was accounting for 50% to 100% of preoperatively measured prostatic adenoma tissue. Matei et al. [19] after reviewing the RASP series reported that the relative amounts of removed tissue was higher, both when mean values (81.2% for RASP versus 75.6% for OSP and 70.4% for TUR) and median values were considered (68.6% for RASP versus 65.7% for OSP and 66.7% for TUR). Jones et al. [57] conducted a systematic review and meta-analysis comparing HoLEP with SP, including OP, RASP, and LSP. Only three randomized studies (263 patients) were included among the evaluated articles. Each trial compared HoLEP with OP. No studies were identified, which compared HoLEP with RASP or LSP directly. The mean prostate volume was 114 mL in the HoLEP group and 119 mL in the simple prostatectomy group. They reported that OSP was associated with a significantly shorter operative time and greater tissue removal. Nevertheless, further studies were still required to reveal unanswered questions about these issues. 

Cost was another controversial topic and should be taken into account. Cost–benefit analysis was a complex problem and affected by multiple factors such as hospital costs, complications, and reimbursement issues which vary significantly between countries and healthcare systems [58]. Sutherland et al. [15] reported that RASP was expectedly expensive compared to OSP, adding an average of USD 2797 to the operating charges, which does not include the initial investment cost of the robotic system. On the other hand, Matei et al. [19] noted that considering the cost of hospitalization, transfusion rates and need for continuous bladder irrigation, RASP may become cheaper overall than OSP and has similar cost to bipolar TUR-P. Pariser et al. [59] found a significant difference in hospital charges related to the presence of complications after SP. While patients with complications cost USD 51,295, patients without a complication cost USD 32,305. Considering that RASP has lower complication rates compared to OSP, one might assume that RASP may cost less overall. Salonia et al. [60] compared the cost of OSP and HoLEP and reported average cost of HoLEP was EUR 2356 which was 9.6% less than OSP. The usage of robotic instruments during RASP differs among surgeons and institutions. Since most instruments were used a maximum of 10 times and each instrument costs USD 2500, using extra arms may increase the cost of RASP. In this regard, the surgeon’s experience and the technique impacts the cost of RASP. Still, more studies are needed to establish the recent and actual cost of RASP. Considering all these cost issues and the longer operation time with RASP, reserving RASP for patients with concomitant bladder stones of considerable size or diverticula, which can be treated simultaneously, is logical [44]. Considering the RASP learning curve, which was at least 10–15 cases, if the caseload of RASP per center is less than 10–15 per year, implementation of RASP should be debated because of aforementioned reasons [44]. Yet, well-designed studies are needed before a clear conclusion can be drawn. 

## 5. Conclusions

For patients with LUTS who need surgery with prostate glands >80 gr, RASP is a good alternative MIS, especially when laser systems or surgical skills are unavailable. This systematic review indicated that RASP does not only provide similar improvements in functional outcomes for IPSS, PVR, Qmax, QoL, but also has similar complication rates, EBL, LOS to OSP, ThuVEP, and HoLEP. While further studies are needed on cost analysis, learning curve, and best surgical approach, implementation of RASP in centers with established robotic programs are becoming attractive and increasing every day. Nevertheless, prospective RCT’s with multicenter and large sample sizes are needed to confirm the results of this systematic review.

## Figures and Tables

**Figure 1 jcm-09-01798-f001:**
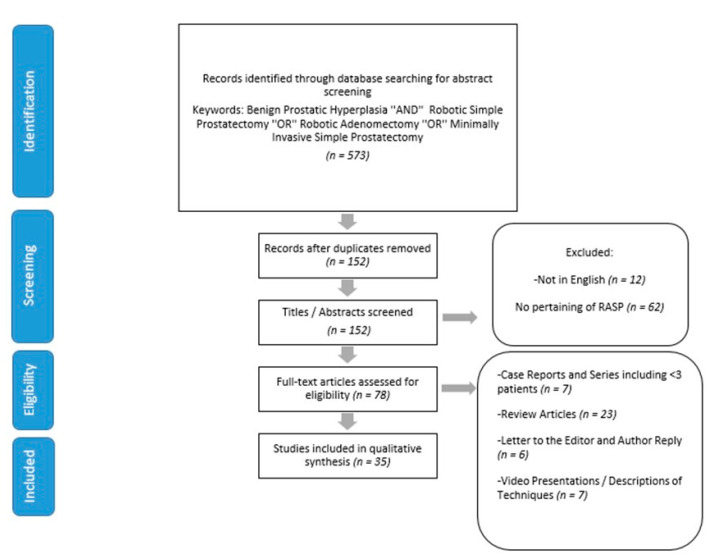
PRISMA 2009 flowchart for the article selection process.

**Table 1 jcm-09-01798-t001:** Robotic-assisted simple prostatectomy (RASP) series in the literature.

Study	Study Type	Year	Number of Patients	Approach	Prostate Volume (mL)	Operation Time (min)	Conversion to OSP (%)	EBL (mL)	Transfusion Rate (%)	Specimen Weight (g)	Catheher Removal Time (Days)	Length of Stay (Days)	Complications (%)	
Sotelo et al. [3]	NC	2008	7	T	77.7	195	0	382	14.3	50.5	7.5	1.3	14	
Yuh et al. [12]	NC	2008	3	T	323	211	0	558	33	301	NA	1.3	33	
John et al. [13]	NC	2009	13	E	100	210	0	500	0	82	6	6	7.7	
Uffort et al. [14]	NC	2010	15	T	70.9	128.8	0	139.3	0	46.4	4.6	2.5	7	
Sutherland et al. [15]	NC	2011	9	T	136.5	183	11.1	206	0	112	13	1.3	56	
Vora et al. [17]	NC	2012	13	T	163	179	0	219	0	127	2.7	8.8	7.7	
Fareed et al. [18]	NC	2012	8	E	130	230	0	425	87.5	78	11	4.5	37.5	
Matei et al. [19]	NC	2012	35	T	96.2	186	0	121	0	87	7.4	3.2	0	
Coelho et al. [16]	NC	2012	6	T	157	90	0	208	0	145	4.8	1	0	
Dubey et al. [35]	NC	2012	3	E	NA	220	0	160	NA	NA	3	3.5	NA	
Clavijo et al. [20]	NC	2013	10	T	81	106	0	375	10	81	8.9	1	20	
Banapour et al. [10]	NC	2014	16	T+E	141.8	228	0	197	0	94.2	8	1.3	12.5	
Leslie et al. [21]	NC	2014	25	T	149.6	214	0	143	4	88	9	4	20	
Stolzenburg et al. [24]	NC	2014	10	E	143.9	122.5	0	228.8	0	102	7.4	8.4	10	
Nestler et al. [22]	NC	2014	18	T	91	193	0	190	5.5	91	5.3	NA	5.5	
Elsamra et al. [23]	NC	2014	15	T	156	189	0	290	0	110	8.67	2.4	14	
Patel et al. [25]	NC	2014	20	T+E	NA	NA	NA	NA	NA	134.7	NA	1.7	NA	
Nething et al. [26]	NC	2014	7	T	144.9	204.7	0	521	NA	96.3	8.28	2.57	0	
Autorino et al. [37]	C	2015	487	T+E	110	154.5	3.1	200	1	75	7	2	16.6	
Pokorny et al. [27]	NC	2015	69	T	129	97	0	200	1.5	84	3	4	33	
Hoy et al. [36]	C	2015	4	T	238	161.3	0	218.8	0	123.6	NA	2.3	0	
Castillo et al. [28]	NC	2016	34	T	117	96	0	200	5.8	76	4.6	2.2	20.5	
Martin Garzon et al. [38]	C	2016	154	E	78	157	0.7	455	6	71	9.3	NA	12	
Pavan et al. [39]	C	2016	130	T+E	118.5	150	10.8	250	NA	77	5	5	16.9	
Umari et al. [40]	C	2017	81	T	130	105	0	250	1.2	89	3	4	31	
Zhang et al. [41]	C	2017	32	E	NA	274	NA	NA	9.4	110	8	2.3	3.1	
Sorokin et al. [42]	C	2017	59	T	136.9	161.4	0	339	3.4	82.9	5.7	1.5	19	
Cacciamani et al. [29]	NC	2018	23	T	108.1	160.6	0	98.6	0	63.1	7	2.1	4.3	
Wang et al. [30]	NC	2018	27	T	82	169	0	235	0	47.5	1.6	3	22.2	
Chavali et al. [31]	NC	2018	28	T	180	180	0	200	0	90	8	2	14	
Johnson et al. [33]	NC	2018	120	T	121.5	157	0	NA	3.3	74	4	1	18.3	
Nestler et al. [44]	C	2019	35	NA	94.5	182	0	NA	9.4	77	5	5	25	
Mourmouris et al. [43]	C	2019	26	T	NA	133.6	0	274	0	115	3	3.4	3.8	
Simone et al. [32]	NC	2019	12	T	102	150	0	250	8	78	7	3	30	
Steinberg et al. [34]	NC	2019	10	E	104	172	0	141	0	65	1.9	1.1	0	
**Study**	**Study Type**	**Preop IPSS**	**Postop IPSS**	**Preop QoL**	**Postop QoL**	**Preop IIEF/SHIM**	**Postop IIEF/SHIM**	**Preop Qmax (mL/sn)**	**Postop Qmax (mL/sn)**	**Preop PVR (mL)**	**Postop PVR (mL)**	**Postop İncontinence**	**Bladder neck Contracture (*n*)**	**Follow up (Months)**
Sotelo et al. [3]	NC	22	1.7	3.8	2.3	NA	NA	17.7	55.5	NA	NA	NA	N/A	0
Yuh et al. [12]	NC	17.7	NA	4.1	NA	NA	NA	NA	NA	NA	NA	0	1	NA
John et al. [13]	NC	NA	NA	NA	NA	NA	NA	NA	23	85	0	NA	1	13
Uffort et al. [14]	NC	23.9	1.8	4.9	2.2	NA	NA	NA	NA	265.8	44.2	NA	N/A	3
Sutherland et al. [15]	NC	17.8	7.8	NA	NA	12.7 (SHIM)	12.5 (SHIM)	NA	NA	214	18.2	1	N/A	9
Vora et al. [17]	NC	18.2	5.3	NA	NA	NA	NA	4.4	19.1	207.3	12.7	0	N/A	7.2
Fareed et al. [18]	NC	19.5	3	NA	NA	NA	NA	5.2	18	60	37	0	N/A	1
Matei et al. [19]	NC	28	7	NA	NA	NA	NA	6.6	18.9	NA	NA	NA	N/A	0
Coelho et al. [16]	NC	19.8	5.5	NA	NA	NA	NA	7.7	19	NA	NA	0	N/A	2
Dubey et al. [35]	NC	NA	NA	NA	NA	NA	NA	NA	NA	NA	NA	NA	N/A	NA
Clavijo et al. [20]	NC	18.8	1.7	3.7	0.5	NA	NA	12.4	33.5	NA	NA	0	N/A	1
Banapour et al. [10]	NC	22	7	4	2	NA	NA	NA	NA	194	56	0	N/A	0
Leslie et al. [21]	NC	23.9	3.6	NA	NA	12.8 (SHIM)	NA	11.3	20	208.1	36.9	1	N/A	6
Stolzenburg et al. [24]	NC	21.9	3.4	NA	NA	NA	NA	9.3	20.7	121.9	57.5	0	N/A	6
Nestler et al. [22]	NC	25	6.1	5	1.1	59 (IIEF)	56.8 (IIEF)	9	28.2	NA	NA	1	0	1
Elsamra et al. [23]	NC	16.2	4.5	NA	NA	NA	NA	NA	NA	428	33	3	N/A	3
Patel et al. [25]	NC	14.7	NA	NA	NA	NA	NA	NA	NA	414	NA	NA	N/A	NA
Nething et al. [26]	NC	NA	NA	NA	NA	NA	NA	NA	NA	NA	NA	0	N/A	10
Autorino et al. [37]	C	23	7	4	NA	15 (SHIM)	15 (SHIM)	8	25	108	NA	NA	3 patients	12
Pokorny et al. [27]	NC	25	3	NA	NA	NA	NA	7	23	73	0	0	N/A	6
Hoy et al. [36]	C	NA	NA	NA	NA	NA	NA	NA	NA	NA	NA	NA	N/A	3
Castillo et al. [28]	NC	23.5	7.1	NA	NA	NA	NA	10.4	23.1	NA	NA	0	1 patient	12
Martin Garzon et al. [38]	C	22	6.5	3.8	1	18.5 (SHIM)	16 (SHIM)	11.5	33	NA	NA	9	1 patient developed anterior urethral stricture	12
Pavan et al. [39]	C	23	5	6	NA	18 (SHIM)	17 (SHIM)	9	22	NA	NA	NA	N/A	10.3
Umari et al. [40]	C	25	5	NA	NA	NA	NA	8	23	73	0	1	0	12
Zhang et al.[41]	C	NA	NA	NA	NA	NA	NA	NA	NA	NA	NA	NA	N/A	NA
Sorokin et al. [42]	C	18.8	7.3	3.9	1.3	NA	NA	9.8	22.4	118	3.5	0	0	6
Cacciamani et al. [29]	NC	23.1	NA	NA	NA	NA	NA	NA	NA	NA	NA	0	N/A	3
Wang et al. [30]	NC	25	NA	6	NA	18 (IIEF)	17.5 (IIEF)	6	NA	85	NA	0	No strictue but 7 patients required closure of small urethrotomies	16.4
Chavali et al. [31]	NC	19	NA	NA	NA	NA	NA	9	NA	120	NA	NA	N/A	NA
Johnson et al. [33]	NC	NA	NA	NA	NA	NA	NA	8.9	18.8	NA	9	0	N/A	3.2
Nestler et al. [44]	C	23	NA	5	NA	NA	NA	NA	NA	NA	NA	0	N/A	12
Mourmouris et al. [43]	C	22.9	5.7	NA	NA	NA	NA	10.1	19.1	178.5	25.5	NA	N/A	3
Simone et al. [32]	NC	33	6	NA	NA	27 (IIEF)	27 (IIEF)	7.7	18.6	175	30	0	N/A	12
Steinberg et al. [34]	NC	20.8	12.9	NA	NA	NA	NA	8.6	11.2	119	40.9	1 (transient)	N/A	28.7

EBL, estimated blood loss; IPSS, international prostate symptom score; QoL, quality of life; Qmax, maximum urinary flow rate; PVR, postvoid residual urine; IIEF, international index of erectile function; SHIM, Sexual Health Inventory for Men; NC, non-comparative; C, comparative.

**Table 2 jcm-09-01798-t002:** RASP versus open simple prostatectomy (OSP) and minimally invasive simple prostatectomy comparative studies.

Reference	Versus	Study Design	Number of Cases	Baseline Characteristics	Main Findings
Pavan et al. [39]	LSP	Multileft retrospective	319 (LSP = 189; RASP = 130)	Median prostate volume larger for RASP (118.5 versus 109 mL; *p* = 0.02)	-No significant difference for blood loss, catheter time, hospital stay, major complication rate -On MVA technique not influencing ‘trifecta’ outcome
Martin Garzon et al. [38]	LSP	Single left retrospective	315 (LSP = 82; IF-RASP = 75)	No differences	-Similar surgical outcomes and functional outcomes at 1 year
Umari et al. [40]	HoLEP	Single left retrospective	126 (HoLEP = 45; RASP = 81)	RASP patients younger (median age 69 versus 74, *p* = 0.032), less healthy (Charlson index >2 in 62% versus 29%, *p* < 0.001), with higher preoperative IPSS (25 versus 21, *p* = 0.049)	-Similar improvement for Qmax, PVR, IPSS -Similar operative time -Catheter time (3 versus 2, *p* = 0.005) and hospital stay (4 versus 2 days, *p* = 0.0001) longer for RASP -Complication rates similar
Zhang et al. [40]	HoLEP	Bileft retrospective	632 (HoLEP = 600; RASP = 32)	No differences	-Mean operative time shorter for HoLEP (103 versus 274 min, *p* < 0.001 -HoLEP with lower transfusion rate (1.8 versus 9.4%, *p* = 0.03), shorter catheter time (0.7 versus 8 days, *p* < 0.001), and shorter hospital stay (1.3 versus 2.3 days, *p* < 0.001) -Complication rates similar
Sorokin et al. [42]	OSP	Single left retrospective propensity score matched	188 (OSP = 59; RASP = 59)	No differences	-RASP with shorter mean hospital stay (1.5 versus 2.6 days, *p* < 0.001), but longer operative time (161 versus 93 min, *p* < 0.001) -Lower blood loss (339 versus 587 mL, *p* < 0.001) and hemoglobin drop (12.3% versus 19.5%, *p* = 0.001) for RASP -No differences in transfusion rates, functional outcomes, complication rate
Mourmouris et al. [43]	OSP	Bileft prospective	41 (OSP = 15; RASP = 26)	RASP patients younger (median age 66.73 versus 70.46 *p* = 0.032),	-RASP achieves similar functional outcomes and provides significant advantages, such as decreased blood loss, faster catheter removal (because of the uneventful postoperative course), a shorter LOS and a lower complication rate, at the cost of a longer operating time
Nestler et al. [44]	OSP, ThuVEP	Multileft, Matched Pair Analysis	105 (OSP = 35; RASP = 35; ThuVEP = 35)	No differences	-Blood loss in OSP was significantly higher compared to the minimal invasive approaches. ThuVEP showed a median operation time of 83 min and was therefore significantly faster than OSP with 130 min (*p* = 0.004) and RASP needing 182 min. Significant advantages for the minimal invasive approaches compared to open surgery concerning blood loss, transfusion rates and early continence
Autorino et al. [37]	LSP	Multileft, Retrospective	1330 (RASP = 487; LSP = 843)	Median Charlson Index for LSP patients: 4 and for RASP patients: 2 Median prostate volume is larger in RASP patients (110 versus 99 mL)	Trifecta outcome, arbitrarily defined as a combination of the following postoperative events: International Prostate Symptom Score <8, maximum flow rate >15 mL/s, and no perioperative complications. Trifecta outcome was not significantly influenced by the type of procedure (robotic versus laparoscopic; *p* = 0.136; odds ratio: 1.6; 95% confidence interval, 0.8–2.9), whereas operative time (*p* = 0.01; OR: 0.9; 95% CI, 0.9–1.0) and estimated blood loss (*p* = 0.03; OR: 0.9; 95% CI, 0.9–1.0) were the only two significant factors.
Hoy et al. [36]	OSP	Single left Retrospective	32 (RASP = 4; OSP = 28)	No differences RASP patients younger (median age 69.3 versus 75.18, *p* = 0.17), Prostate volume on TRUS (mL); RASP = 239 ± 49.8 OSP = 180 ± 54.7 0.09	-There was a significant difference in the mean length of operation, with RASP exceeding OSP (161 versus 79 min; *p* = 0.008). -The mean intraoperative blood loss was significantly higher in the open group (835.7 versus 218.8 mL; *p* = 0.0001). -Mean LOS was shorter in the RASP group (2.3 versus 5.5 days; *p* = 0.0001). -No significant differences were noted in the 90-day transfusion rate (*p* = 0.13), or overall complication rate at 0% with RASP versus 57.1% with OSP (*p* = 0.10).

HoLEP, Holmium laser enucleation of the prostate, ThuVEP: Thulium laser VapoEnucleation, IF-RASP, intrafascial robotic assisted simple prostatectomy, LSP, laparoscopic simple prostatectomy, MVA, multivariable analysis, OSP, open simple prostatectomy, PVR, postvoid residual.

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
