# Peer review of "Robotic-Assisted Simple Prostatectomy: A Systematic Review"

_jcm, 2020, doi:10.3390/jcm9061798_

Round 1

Reviewer 1 Report

Authors need to re-review the language of the manuscript, to eradicate a few errors. This will help improve the read flow. 

Author Response

Dear Reviewer,

Thank you very much for your suggestions. We have reviewed the language of the manuscript and corrected some grammatical errors. You can easily tract the changes made in the re-submitted manuscript.  I hope these changes will help improve the read flow. With our best regards,

Reviewer 2 Report

The authors performed the systematic review of eligible literatures on RASP, and they concluded that RASP is a viable MIS treatment option for patients with male LUTS needing surgical treatment for enlarged prostates. They accomplished the very extensive work.

Major comments:
Introduction: adequate

Methods: adequate but not mentioning the registration of study protocol (PROSPERO #?)

Results: Readers' readability is worrisome because it is too narrative. It would be nice to divide it into several sections.

Discussion: adequate

Conclusion:
They concluded that for patients with LUTS who need surgery with postate glands> 80 gr, RASP is a good alternative MIS, but this sentence seems to be too strong. At this point, I think that it is true only when laser system or surgical skills are unavailable in the current situation.

Minor comments:
There are many grammatical errors as belows.
line # 40 (> 80) gr>> 80 gr
line # 41 Open Simple Prostatectomy> open simple prostatectomy
line # 78 RT> randomized clinical trial (RCT)
line # 119 eligable> eligible
ThuVEP or ThuLEP? Different modalities? Couldn't you choose either?

Author Response

To Reviewer 2,

Thank you very much for your comments. We reviewed the manuscript and re-wrote the sections according to your suggestions. You can easily tract the changes from the re-submitted manuscript.

  1. Regarding the PROSPERO number, this study was not registered as such. We are happy to still do so, but, the web site states that the study should not be completed to be eligible for PROSPERO registration. Besides, if we register this study as a new study it will take some time for processing to get a PROSPERO number. This manuscript was prepared for BPH special issue and it should be kept in mind that this may significantly further delay the  publication. We feel very sorry about that this was not brought to our attention earlier.
  2. We have divided results section into different paragraphs so that a reader can follow results separately in categories.
  3. Conclusion  is corrected according to your suggestion.
  4.  Many grammatical errors are corrected as well.
  5. ThuLEP was replaced with ThuVEP all across the manuscript. 

We will be very happy to make revisions if you have additional comments. With our best regards

This manuscript is a resubmission of an earlier submission. The following is a list of the peer review reports and author responses from that submission.

Round 1

Reviewer 1 Report

Authors have done a good job in summarizing and comparing the outcomes of robotic simple prostatectomy with other modalities specially open technique. 

I would like to know if BNC has been reported as a complication in any of the RSP, and rates compared with OSP. 

Can authors shed light on the need of RCT or not, based on their current findings. 

A few language errors are noted that need corrections. 

Reviewer 2 Report

Good review. Good concept.

I think the Introduction section needs to be trimmed significantly.

The Discussion also needs to be more focused - quite repetitive at times.

Some minor English language editing is also required.

As such, the authors need to either provide more detailed methodology on their search strategy...or, they need to amend the Title.

Major improvements in the Introduction and Discussion section are required.

Reviewer 3 Report

The paper presents a systematic review of robotic-assisted simple prostatectomy (RASP) of case series and case studieswith three or more patients. The paper presents recent clinical data in RASP and other techniques such as OSP or other MIS techniques(HoLEP, ThuLEP, etc.). The paper reviews RASP procedures from the first published procedure in 2008 until today. The paper has identified new comparative studies which add value to the systematic review. The nonexistence of comparative studies what was one of the main conclusions from  a previous review papers published for RASP by Banapour et al.

The paper is very well written and structured. It addresses the most important  questions and concerns with respect to RASP and comparing it to other MIS techniques.  

The English language is excellent with small grammatical errors, here are some of the errors (not all): line56 “have been emerged”, line64 “outcomes has improved”, line65 “with the time”, line122 “lunguages”, “el al” should be written as “el al.” throughout the paper, line162-Table2-ref[44] “SSignificant”, line261 “lesser” -> “lower”, line262 “use” -> “used”, line263 “have 10 lives” -> use better English – instruments do not have “lives”, line264 “impacts on cost” -> “impacts the cost”.

Some more detailed comments on the paper:

Line 93 – Are there other systematic review papers about RASP and what is the main contribution of your paper?

Line96 – METHODS -The search terms in the Search Strategy sub-section and in the “Identification” in the Figure 1 is not the same. Which search term is correct? Please correct.

Figure 1 – “Eligibility” – write the number of articles that were excluded in each of the three classifications: “Case reports”, “ Case series <3”, “Review articles”

Table 1:

Line 122 – You state that you have found 33 eligible studies. 23 case and 10 comparative studies. All these studies should be in table 1. But Table 1 is missing references [25], [31], [37]. Furthermore you have many mistakes with references in Table 1 – the reference [7] is in duplicate Castilo et al. and Autorino et al.? Reference [44] is in duplicate Nestler et al.? Wang et al. and Chavali et al. do not have reference numbers. You should correct all the references considering the large number of errors.

Furthermore  Table 1 is hardly comprehendible because it has too many columns. Try splitting the table in two tables. Furthermore because you have ordered the papers in Table 1 by year it is hard to differentiate the non-comparative with comparative studies – please add a column describing this.

Lastly, Table 1 has only 32 lines and in the text you are mentioning  33 references: [3,4,15-35] [7,36-44], but one is missing from the table. Please correct.

Why was the paper of Johnson et al. (doi:10.1089/end.2018.0377) not included in the literature review? This paper provides a retrospective review of RASP on 120 consecutive cases performed by two experienced robotic surgeons from 2014 to 2017. The paper fully fits within the scope of this review paper and can be found using these search terms: “Benign prostatic hyperplasia” and “robotic simple prostatectomy” on Pubmed. Include this paper in the review.

Discussion:

More information concerning the operative time should be given. For example why is the mean operative time of Pokory et al. [30] under 100 minutes which is directly comparable to LSP and all other approaches. Umari et al. also has an operative time of just 105 minutes in a series of 81 patients. Is there some difference between the robotic systems used? Give more information why is the mean operative time with respect to other studies much shorter in some studies.

Furthermore there is no mention of the robot manufacturer for none of the studies. Probably most of the used robots are the DaVinci but many other manufacturers exist on the market today. Is there some difference between the operative time or some other results of the operation concerning the robot used?

Line 254 – Cost – The cited papers from Sutherland et al. and Matei et al. are paper from 2011 and 2012. From that time 8 years have passed and one of the major breakthroughs in robotic technology is also its availability and the price which is  decreasing (or new products on the market are announced that are cheaper). Taking this into account some newer data concerning the cost of RASP should be given. Furthermore the information that was given by Matei et al. and Sutherland et al. are also available in the book chapter from de Carvalho et al. (10.1007/978-3-319-64704-3_10) – maybe this reference and its findings should also be cited in the discussion.

Line 256 – what are the mean investments into the robotic technologies – there are multiple players on the market so a price range should be added.

Line 267 – The last sentence of this paragraph in ambiguous: “If the case load per center Is less than 10-15…”: in what time frame – per year/month? Why and how should it be debated? Please make this sentence clearer.

Conclusion:

“While further studies are needed on cost analysis, learning curve and best surgical approach…”. In the paper by Johnson et al., as previously denoted, the authors have made a retrospective review of RASP on 120 consecutive cases performed by two experienced robotic surgeons from 2014 to 2017. Furthermore they have  defined the learning curve for RASP and made a detailed overview and discussion concerning the ‘‘learning curve’’. Add the findings of  these authors to your paper because they give a very good overview of the learning curve.

All abbreviations should be defined on their first mention in the manuscript.

References

Johnson B, Sorokin I, Singla N, Roehrborn C, Gahan JC. Determining the Learning Curve for Robot-Assisted Simple Prostatectomy in Surgeons Familiar with Robotic Surgery. Journal of Endourology. 2018;32(9):865-870. doi:10.1089/end.2018.0377

de Carvalho PA, Coelho RF. Surgical Treatment: Robotic Simple Prostatectomy. In: Kasivisvanathan V, Challacombe B, eds. The Big Prostate. Cham: Springer International Publishing; 2018:129-142. doi:10.1007/978-3-319-64704-3_10

Reviewer 4 Report

The authors systematically reviewed contemporary minimally invasive surgical treatment options of patients with male LUTS in men with prostate glands > 80 mL including HoLEP, ThuVEP and LSP or RASP. They mainly focused on RASP, and concluded that RASP provided similar improvements in functional outcomes and had similar complication rates, EBL, and LOS to OSP, ThuVEP and HoLEP.

This manuscript was well-organized and concisely-written. But there were several errors in spelling and editing (use of abbreviation). These should be extensively revised.

I also recommend to summarize the important finding of the former systematic reviews and the relevant meta-analysis (Li et al. and Leonardo et al.), and to introduce RASP using SP flatform (Steinberg et al).

Comparison Between Minimally Invasive Simple Prostatectomy and Open Simple Prostatectomy for Large Prostates: A Systematic Review and Meta-Analysis of Comparative Trials.

Li J, Cao D, Peng L, Ren Z, Gou H, Li Y, Wei Q.

J Endourol. 2019 Sep;33(9):767-776. doi: 10.1089/end.2019.0306. Epub 2019 Jul 26.

What is the standard surgical approach to large volume BPE? Systematic review of existing randomized clinical trials.

Leonardo C, Lombardo R, Cindolo L, Antonelli A, Greco F, Porreca A, Veneziano D, Pastore A, Dalpiaz O, Ceruti C, Verze P, Borghesi M, Schiavina R, Falabella R, Minervini A; AGILE Group.

Minerva Urol Nefrol. 2020 Feb;72(1):22-29. doi: 10.23736/S0393-2249.19.03589-6.

Initial experience with extraperitoneal robotic-assisted simple prostatectomy using the da Vinci SP surgical system.

Steinberg RL, Passoni N, Garbens A, Johnson BA, Gahan JC.

J Robot Surg. 2019 Sep 27. doi: 10.1007/s11701-019-01029-7